# Learning to control the pitch of speech signals in the latent representation of a variational autoencoder

## Abstract

Understanding and controlling latent representations in deep generative models is a challenging yet important problem for analyzing, transforming and generating various types of data. Speech signals are produced from a few physically meaningful continuous latent factors governed by the anatomical mechanisms of phonation. Among these factors, the fundamental frequency is of primary importance as it characterizes the pitch of the voice, which is an important feature of the prosody. In this work, from a variational autoencoder (VAE) trained in an unsupervised fashion on hours of natural speech signals and a few seconds of labeled speech generated with an artificial synthesizer, we propose a weakly-supervised method to (i) identify the latent subspace of the VAE that only encodes the pitch and (ii) learn how to move into this subspace so as to precisely control the fundamental frequency. The proposed approach is applied to the transformation of speech signals and compared experimentally with traditional signal processing methods and a VAE baseline.

## 1 Introduction

The variational autoencoder (VAE) (Kingma & Welling, 2014; Rezende et al., 2014), which is equipped with both a generative and inference model, can be used not only for data generation but also for analysis and transformation. Making sense of the latent representation learned by a VAE and controlling the underlying continuous factors of variation in the data are important challenges to build more expressive and interpretable generative models.

Variants of the VAE have recently led to considerable progress in disentangled representation learning (Kim & Mnih, 2018; Higgins et al., 2017; Chen et al., 2018; Locatello et al., 2019; 2020). This is a first step towards the control of the high-level generative properties of the data. Several methods have been recently proposed to control continuous factors of variation in deep generative models (Jahanian et al., 2019; Plumerault et al., 2020; Goetschalckx et al., 2019; Härkönen et al., 2020), focusing essentially on generative adversarial networks (Goodfellow et al., 2014). They consist in identifying and moving onto semantically meaningful directions in the latent space of the model. These previous works however only allow for moving "blindly" in the latent space. Being able to control the data generation process conditioned on a specific value for a given factor of variation remains a difficult problem in deep generative modeling, that we propose to address in this paper.

Previous works on representation learning with deep generative models, in particular VAEs, have mostly focused on images (Higgins et al., 2017; Chen et al., 2018; Kim & Mnih, 2018). Yet, it is not always easy to define the ground-truth latent factors of variation involved in the generation of natural images. For speech data, the latent factors of variation can be directly related to the anatomical mechanisms of speech production. This makes speech data interesting for investigating the disentangled representation learning capabilities of VAEs, complementary to studies dealing with images. The seminal work of Hsu et al. (2017) proposed to modify the speaker identity and the phonemic content of speech signals using translations in the latent space of a VAE. However, this method requires to know predefined values of the latent representation associated not only with the output but also with the input speech attributes to be modified. Moreover, Hsu et al. (2017) did not address the control of continuous factors of speech variation in the VAE latent space.

A key concept for characterizing the structure of speech signals is deduced from the source-filter model proposed by Fant (1970). This model implies that a speech signal is mainly characterized by a few continuous latent factors of variation corresponding to the vibration of the vocal folds (i.e., the source), which defines the fundamental frequency, and the resonances of the vocal tract (i.e., the filter), which define the formants. In the present study, we first train a VAE on a dataset of about 25 hours of unlabeled speech signals. Then, using only a few seconds of automatically-labeled speech signals generated with an artificial speech synthesizer, we propose a method to analyze and control the fundamental frequency in the latent representation of the previously trained VAE. We introduce a general formalism based on the use of a "biased" aggregated posterior, from which we can identify a latent subspace encoding the factor of variation to be controlled. We then propose a method to learn to move into this latent subspace, so as to precisely control the corresponding factor while leaving other characteristics (e.g., the speech formants) unchanged. The proposed method is applied to the modification of the fundamental frequency of speech signals.

## 2 LEARNING TO CONTROL A CONTINUOUS LATENT FACTOR OF VARIATION

Let $\mathbf{x} \in \mathbb{R}^D$ denote the observed high-dimensional data vector. In the present paper, it corresponds to the power spectrum of a short frame of speech signal (i.e., a column of the short-time Fourier transform (STFT) power spectrogram). Its entries are thus non negative and its dimension $D$ equals the number of frequency bins. We use the Itakura-Saito VAE defined by

$$p(\mathbf{z}) = \mathcal{N}(\mathbf{z}; \mathbf{0}, \mathbf{I}), \qquad p_\theta(\mathbf{x}|\mathbf{z}) = \prod_{d=1}^{D} \mathrm{Exp}\left([\mathbf{x}]_d; [\mathbf{v}_\theta(\mathbf{z})]_d^{-1}\right), \tag{1}$$

where $\mathcal{N}$ and Exp denote the densities of the multivariate Gaussian and univariate exponential distributions, respectively, and $[\mathbf{v}]_d$ denotes the $d$-th entry of $\mathbf{v}$ (Bando et al., 2018; Leglaive et al., 2018; Girin et al., 2019). The inverse scale parameters of $p_\theta(\mathbf{x}|\mathbf{z})$ are provided by a decoder neural network, parametrized by $\theta$ and taking $\mathbf{z} \in \mathbb{R}^L$ as input, $L \ll D$. We also introduce an inference model $q_\phi(\mathbf{z}|\mathbf{x}) \approx p_\theta(\mathbf{z}|\mathbf{x})$ defined by

$$q_\phi(\mathbf{z}|\mathbf{x}) = \mathcal{N}\left(\mathbf{z}; \boldsymbol{\mu}_\phi(\mathbf{x}), \mathrm{diag}\{\mathbf{v}_\phi(\mathbf{x})\}\right), \tag{2}$$

where the mean and variance parameters are provided by an encoder neural network, parametrized by $\phi$ and taking $\mathbf{x}$ as input (Kingma & Welling, 2014; Rezende et al., 2014). The VAE training consists in maximizing a lower-bound of $\ln p_\theta(\mathbf{x})$, called the evidence lower-bound (ELBO) and defined by $\mathcal{L}(\theta, \phi) = \mathbb{E}_{\hat{p}(\mathbf{x})}\left[\mathbb{E}_{q_\phi(\mathbf{z}|\mathbf{x})}[p_\theta(\mathbf{x}|\mathbf{z})] - D_{\mathrm{KL}}\left(q_\phi(\mathbf{z}|\mathbf{x}) \parallel p(\mathbf{z})\right)\right]$ (Neal & Hinton, 1998).

### 2.1 LEARNING A LATENT SUBSPACE ENCODING THE FACTOR OF VARIATION

Let $\mathcal{D}$ denote a small dataset of artificially-generated data vectors $\mathbf{x}$ in which one single factor of variation $f_0$ changes. In this paper, $\mathcal{D}$ corresponds to short-term speech power spectra and $f_0$ denotes the fundamental frequency, all other factors (loudness, formants, ...) are arbitrarily fixed. Because only $f_0$ varies in $\mathcal{D}$, we expect latent vectors drawn from the "biased" aggregated posterior, $\hat{q}_\phi(\mathbf{z}) = \frac{1}{\#\mathcal{D}} \sum_{\mathbf{x}_n \in \mathcal{D}} q_\phi(\mathbf{z}|\mathbf{x}_n)$, to live on a lower-dimensional manifold embedded in the original latent space $\mathbb{R}^L$. This defines the explicit inductive bias (Locatello et al., 2019) that we propose to exploit to learn the latent fundamental frequency representation of speech in the VAE. We assume this manifold to be a subspace characterized by its semi-orthogonal basis matrix $\mathbf{U} \in \mathbb{R}^{L \times M}$, $1 \leq M < L$. This matrix is computed by solving the following optimization problem:[1]

$$\min_{\mathbf{U} \in \mathbb{R}^{L \times M_i}} \mathbb{E}_{\hat{q}_\phi(\mathbf{z})}\left[\left\|\mathbf{z} - \mathbf{U}\mathbf{U}^\top\mathbf{z}\right\|_2^2\right], \qquad s.t. \ \mathbf{U}^\top\mathbf{U} = \mathbf{I}. \tag{3}$$

The space spanned by the columns of $\mathbf{U}$ is a subspace of the original latent space $\mathbb{R}^L$ in which the latent vectors associated with the variation of the factor $f_0$ in $\mathcal{D}$ are expected to live. We can show that, similarly to principal component analysis (PCA) (Pearson, 1901), the solution to the optimization problem (3) is given by the $M$ eigenvectors corresponding to the $M$ largest eigenvalues of $\mathbf{S}_\phi(\mathcal{D}) = \frac{1}{\#\mathcal{D}} \sum_{\mathbf{x}_n \in \mathcal{D}} \mathbb{E}_{q_\phi(\mathbf{z}|\mathbf{x}_n)}[\mathbf{z}\mathbf{z}^\top]$, which can be computed analytically from the encoder outputs. The dimension $M$ of the subspace can be chosen such as to retain a certain percentage of the data variance in the latent space.

---

[1]In the rest of the paper, without loss of generality, we assume that, for each data vector in $\mathcal{D}$, the associated latent vector $\mathbf{z}$ has been centered by subtracting $\boldsymbol{\mu}_\phi(\mathcal{D}) = \mathbb{E}_{\hat{q}_\phi(\mathbf{z})}[\mathbf{z}] = \frac{1}{\#\mathcal{D}} \sum_{\mathbf{x}_n \in \mathcal{D}} \boldsymbol{\mu}_\phi(\mathbf{x}_n)$.

## 2.2 Controlling the factor of variation in the learned latent subspace

We have defined a methodology to learn a latent subspace $\mathbf{U} \in \mathbb{R}^{L \times M}$ that encodes $f_0$ in the dataset $\mathcal{D}$, containing for instance a few examples of automatically-labeled speech data generated by an artificial synthesizer. Making now use of the labels in $\mathcal{D}$, we propose to learn a regression model $\mathbf{g}_\eta : \mathbb{R}_+ \mapsto \mathbb{R}^M$ from the factor $f_0$, whose value is denoted by $y \in \mathbb{R}_+$, to the data coordinates in the latent subspace defined by $\mathbf{U}$. The parameters $\eta$ are estimated by solving

$$\min_\eta \left\{ \mathbb{E}_{\hat{q}_\phi(\mathbf{z}, y)} \left[ \left\| \mathbf{g}_\eta(y) - \mathbf{U}^\top \mathbf{z} \right\|_2^2 \right] \stackrel{c}{=} \frac{1}{\#\mathcal{D}} \sum_{(\mathbf{x}_n, y_n) \in \mathcal{D}} \left\| \mathbf{g}_\eta(y_n) - \mathbf{U}^\top \left( \boldsymbol{\mu}_\phi(\mathbf{x}_n) - \boldsymbol{\mu}_\phi(\mathcal{D}) \right) \right\|_2^2 \right\},$$
(4)

where $\hat{q}_\phi(\mathbf{z}, y) = \int q_\phi(\mathbf{z}|\mathbf{x}) \hat{p}(\mathbf{x}, y) d\mathbf{x}$, $\hat{p}(\mathbf{x}, y)$ is the empirical distribution associated with $\mathcal{D}$, considering now both the data vector $\mathbf{x}$ and the label $y$, and $\stackrel{c}{=}$ denotes equality up to an additive constant w.r.t. $\eta$. The dataset $\mathcal{D}$ is expected to be orders of magnitude smaller than the dataset used to trained the VAE, and as it is synthetic and labels are not provided by human annotators, the problem can be considered very weakly supervised.

We can now transform a speech spectrogram by analyzing it with the VAE encoder, then linearly moving in the learned subspace using the above regression model, and finally resynthesizing with the VAE decoder. Given a source latent vector $\mathbf{z}$ and a target value $y$ for the factor $f_0$, we apply the following affine transformation:

$$\tilde{\mathbf{z}} = \mathbf{z} - \mathbf{U}\mathbf{U}^\top \mathbf{z} + \mathbf{U}\mathbf{g}_\eta(y).$$
(5)

This transformation consists in (i) subtracting the projection of $\mathbf{z}$ onto the subspace associated with the factor of variation $f_0$; and (ii) adding the target component provided by the regression model $\mathbf{g}_\eta$ mapped from the learned subspace to the original latent space by the matrix $\mathbf{U}$. This operation allows us to move only in the latent subspace associated with the factor $f_0$, hopefully leaving other data characteristics unchanged. This approach contrasts with the common one consisting in performing transformations (e.g. interpolations) in the whole VAE latent space, for example as in Hsu et al. (2017) for speech transformations. Note that equation (5) also defines a generative model *conditioned* on the value $y$ of the factor $f_0$, if $\mathbf{z}$ is sampled from the prior $p(\mathbf{z})$ in equation (1) and $\tilde{\mathbf{z}}$ is then mapped through the VAE decoder.

## 3 Experiments

The VAE is trained on about 25 hours of multi-speaker speech data from the Wall Street Journal (WSJ0) dataset (Garofolo et al., 1993). The encoder and decoder networks are simple perceptrons with 2 hidden layers. The data and latent space dimensions are 513 and 16. The proposed method uses only 3.6 seconds of labeled synthetic speech data, which is several orders of magnitude lower than the amount of unlabeled data used to train the VAE model. Using the artificial speech synthesizer Soundgen (Anikin, 2019), we generate a trajectory of 226 speech spectra where $f_0$ varies linearly in $[85, 310]$ Hz, all other parameters of the synthesizer being arbitrarily fixed. When solving the optimization problem (3), the latent subspace dimension $M = 4$ is chosen such that 80% of the data variance is retained. The regression model in Section 2.2 is piecewise linear. Given a transformed spectrogram, we use Waveglow (Prenger et al., 2019) to reconstruct the waveform.

**Qualitative results**   In Figure 1, from left to right we show the original spectrogram of a speech signal uttered by a female speaker (left) and the transformed spectrograms where the pitch is modulated (middle) or has been removed (right), i.e., the original voiced speech signal is converted into a whispered speech signal . This last spectrogram is simply obtained by subtracting to $\mathbf{z}$ its projection onto the latent subspace corresponding to $f_0$ (i.e., by considering only the two first terms in the right-hand side of Equation (5)). It results in a spectrogram where the harmonic component is neutralized, while preserving the original formant resonances structure, originating from the shape of the vocal tract. This is remarkable considering that the VAE was not trained on whispered speech signals. Other qualitative results can be found at `https://tinyurl.com/iclr2022`.

**Quantitative results**   We use a corpus of 12 English vowels uttered by 50 male and 50 female speakers (Hillenbrand et al., 1995). This dataset is labeled with the fundamental and formant frequencies. We apply transformations of the fundamental frequency targeting values in the range

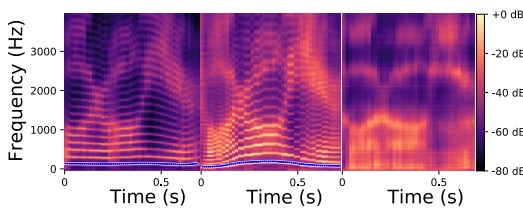

Figure 1: $f_0$ transformation (middle and right) of a speech signal uttered by a female speaker (left).

Table 1: Results of fundamental frequency transformation of English vowels.

| Method | NISQA (↑) | $\delta f_0$ (%,↓) | $\delta f_1$ (%,↓) | $\delta f_2$ (%,↓) |
|---|---|---|---|---|
| TD-PSOLA | 2.32 ± 0.55 | 3.8 ± 2.5 | 6.3 ± 2.8 | 3.7 ± 0.9 |
| WORLD | 2.49 ± 0.60 | 4.5 ± 0.6 | 3.7 ± 1.8 | 2.3 ± 0.7 |
| VAE baseline | 1.94 ± 0.43 | 6.2 ± 2.8 | 10.4 ± 2.4 | 6.2 ± 0.9 |
| Proposed | 2.08 ± 0.48 | 0.8 ± 0.2 | 7.2 ± 1.3 | 3.6 ± 1.2 |

$[100, 300]$ Hz with a step of 1 Hz. We measure the performance regarding three aspects: First, in terms of *accuracy* by comparing the target value ($y$ in equation (5)) and its estimation computed from the modified output speech signal. Second, in terms of *disentanglement*, by comparing the values of the formant frequency $f_j$ for $j \neq 0$, before and after modification of the factor $f_0$. Third, in terms of speech *naturalness* of the transformed signal. Accuracy and disentanglement are measured in terms of relative absolute error (in percent, the lower the better). $\delta f_0$ measures the accuracy of the transformation on $f_0$ while $\delta f_1$ and $\delta f_2$ are used to assess if the formants $f_1$ and $f_2$ remained unchanged after modifying $f_0$. We use CREPE (Kim et al., 2018) to estimate the fundamental frequency and Parselmouth (Jadoul et al., 2018), which is based on PRAAT, to estimate the formant frequencies. Regarding speech naturalness, we use the objective measure provided by NISQA (the higher the better) (Mittag & Möller, 2020). As a reference, the score provided by NISQA on the original dataset of English vowels (i.e., without any processing) is equal to $2.60 \pm 0.53$.

We compare the proposed approach with several methods from the literature: (i) Time-domain pitch-synchronous overlap-and-add (TD-PSOLA) (Moulines & Charpentier, 1990); (ii) WORLD vocoder (Morise et al., 2016); (iii) The method proposed by Hsu et al. (2017) (here referred to as "VAE baseline"), which applies translations directly in the latent space of the VAE. All the methods we compare with require a pre-processing of the input speech signal to compute the input trajectory of the fundamental frequency, which is not the case of the proposed method.

Experimental results (mean and standard deviation) are shown in Table 1. Compared to the VAE baseline, the proposed method obtains better performance in terms of accuracy, disentanglement, and naturalness. These results confirm the effectiveness of performing the transformation in the learned subspace instead of directly in the VAE latent space, as well as the advantage of using regression models instead of predefined latent attribute representations. WORLD obtains the best performance in terms of disentanglement, which is because the pitch and formant contributions are decoupled in the architecture of the vocoder. In terms of naturalness, WORLD and then TD-PSOLA obtain the best performance. This is explained by the fact that these method operate directly in the time domain, therefore they do not suffer from phase reconstruction artifacts, unlike the proposed and VAE baseline methods. We want to emphasize that the objective of this study is not to compete with traditional signal processing methods specifically designed for the task at hand. It is rather to advance on the understanding of deep generative modeling, and to compare honestly with highly-specialized traditional systems when applied to speech signals.

## 4 CONCLUSION

In this work, using only a few examples of artificially generated labeled data, we proposed a weakly-supervised method to control a target factor of variation within a VAE latent space. The effectiveness of the method was illustrated for controlling the fundamental frequency of speech signals. We have applied the same methodology in order to also control the speech formants, and the results are conclusive. In particular, the learned latent subspaces corresponding to the pitch and formants are orthogonal (which can be simply measured from the learned matrices $\mathbf{U}$), demonstrating the disentanglement of the learned representations. To the best of our knowledge, this is the first approach that, with a single methodology, is able to extract, identify and control the source and filter low-level speech attributes within a VAE latent space. The proposed method could be applied to other types of data such as natural or face images, provided that one can create a few synthetic images that capture variations in a single latent factor of interest at a time, independently of others.

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
