# OpenReview forum: "Learning to control the pitch of speech signals in the latent representation of a variational autoencoder"
_ICLR.cc/2022/Workshop/DGM4HSD — Submitted to ICLR 2022 DGM4HSD workshop_

### Official Review · Reviewer_7yuA · 2022-03-24
**Inadequate results and missing generalization to multivariate control**

**Rating:** 3
**Confidence:** 4

**Review:**

In this work, the authors present a novel method to control a continuous property/statistic (y) of samples of a deep generative model.  The weakly supervised technique is designed to learn a latent representation through unsupervised learning by maximizing the ELBO of a VAE on a large dataset, and then using a small amount of supervised data to learn a continuous map from y to a subspace of the latent variable.  In the submission, the authors apply the technique to fundamental frequency adaption of speech.  Furthermore, they claim that the technique can be generalized to multivariate continuous control of factors of variation.

The method as applied in the paper is described well.  The architecture, optimization, and data are clearly described.  I do question the use of an MLP rather than a conv net as is standard in the literature they rerference. The authors claim to have applied this method to multivariate y, however, the exact manner in which this is done is not described. Is the latent space partitioned for each variable in y, or are all variables in y shared in one subspace?

The preliminary results presented are inadequate for this workshop. The authors produce qualitative results (adapted spectrograms) and audio samples of speech with adapted fundamental frequency.  The whispering samples are intriguing considering you don’t need to estimate the fundamental frequency to produce them.  However, the whispers sound quite unnatural likely due to the significant formant distortion (next paragraph).

The quantitative result that this method allows the best control of f_0 is trivial since the compared methods are not directly optimizing control of f_0 as the presented method is.  The comparisons to SOTA techniques in terms of natural speech and formant distortion are appreciated, however they highlight the fact that the formants are being distorted greatly, which affects the quality of the reconstructed (post adaption) speech.

The presented method is rather practical and unoriginal, and does not offer significant insight from the perspective of machine learning research. I consider the treatment of multivariate y to be fundamental to this work, and that is missing entirely.  I recommend that the authors first address the open questions in how to achieve multivariate control on a toy dataset (2D faces, sprites etc) via proof of concept, and then graduate to more sophisticated problems like speech modeling.

While the paper is well-written, the technique presented is well-motivated, and the authors sufficiently reference research on disentangling, this work lacks substantive results (either qualitative or quantitative) that are required for acceptance at this workshop.  The value this work may have provided through the lens of methodological novelty is missing in its incomplete description of how to generalize this technique beyond univariate y.  For these reasons, I recommend rejecting this submission.

---

### Official Review · Reviewer_jXNh · 2022-03-29
**good match to this workshop**

**Rating:** 6
**Confidence:** 3

**Review:**

This paper proposes a VAE to learn a low dimensional interpretable latent representation of speech. By manipulating this representation, the authors are able to control the pitch of the speech. The paper is well written, and it is an interesting application of using deep generative models to learn the structure of a data source. The presented results seem somwhat preliminary and do not obviously outperform alternative methods, but empirical support is sufficient for a proof of concept / workshop paper.

---

### Decision · Program_Chairs · 2022-03-27

**Decision:**

Reject

**Comment:**

The reviewer agrees that the paper is well-motivated and well-written. However, concerns about the missing of quantitatively and qualitatively substantive results have been raised. The AC encourages the authors to take into account the review's suggestions and resubmit to a future venue.